# Prevalence and transmission of the most relevant zoonotic and vector-borne pathogens in the Yucatan peninsula: A review

**Ma. Fernanda Sánchez-Soto**[1]\*, **Osiris Gaona**[1], **Ana Laura Vigueras-Galván**[2,3],
**Gerardo Suzán**[3,4], **Luisa I. Falcón**[1]\*, **Ella Vázquez-Domínguez**[⑤][5]\*

**1** Laboratorio de Ecología Bacteriana, Instituto de Ecología, Unidad Mérida, Universidad Nacional Autónoma de México, Yucatán, México, **2** Laboratorio de Virología, Departamento de Microbiología e Inmunología, Facultad de Medicina Veterinaria y Zootecnia, Universidad Nacional Autónoma de México, Ciudad de México, México, **3** International Joint Laboratory Ecosystem, biological diversity, habitat modifications, and risk of emerging pathogens and diseases in Mexico (ELDORADO), UNAM-IRD, Mérida, México, **4** Laboratorio de Ecología de Enfermedades y Una Salud, Departamento de Etología, Fauna Silvestre y Animales de Laboratorio, Facultad de Medicina Veterinaria y Zootecnia, Universidad Nacional Autónoma de México, Ciudad de México, México, **5** Laboratorio de Genética y Ecología, Departamento de Ecología de la Biodiversidad, Instituto de Ecología, Universidad Nacional Autónoma de México, Ciudad de México, México

\* maria.sanchez@iecologia.unam.mx (MFSS); falcon@ecologia.unam.mx (LIF); evazquez@ecologia.unam.mx (EVD)

## Abstract

### Background

Habitat modification and land use changes impact ecological interactions and alter the relationships between humans and nature. Mexico has experienced significant landscape modifications at the local and regional scales, with negative effects on forest cover and biological biodiversity, especially in the Yucatan peninsula in southeastern Mexico. Given the close relationship between landscape modification and the transmission of zoonotic and vector-borne diseases, it is essential to develop criteria for identifying priority zoonoses in the south of the country.

### Methodology/Principal findings

We reviewed 165 published studies on zoonotic and vector-borne diseases in the region (2015–2024). We identified the most frequent vectors, reservoirs, and hosts, the most prevalent infections, and the factors associated with transmission risk and the anthropogenic landscape modification in urban, rural, ecotone, and sylvatic habitats. The most relevant pathogens of zoonotic risk included *Trypanosoma cruzi*, arboviruses, *Leishmania*, *Rickettsia*, *Leptospira*, and *Toxoplasma gondii*. *Trypanosoma cruzi* was the vector-borne agent with the largest number of infected vertebrate species across habitats, while *Leishmania* and arboviruses were the ones that affected the greatest number of people. Dogs, cats, backyard animals, and their hematophagous ectoparasites are the most likely species maintaining the transmission cycles in human settlements, while rodents, opossums, bats, and other synanthropic animals facilitate connection and transmission cycles between forested habitats with human-modified landscapes. Pathogens displayed different prevalences

**Data Availability Statement:** All relevant data are within the manuscript and in the Supporting information file.

**Funding:** This study received financial support from Programa de Apoyo a Proyectos de Investigación e Innovación Tecnológica-Dirección General de Asuntos del Personal Académico (PAPIIT-DGAPA, UNAM; project No. IV200421) awarded to LIF, EVD, and GS, including a Postdoctoral scholarship from DGAPA to MFSS. The funders had no role in study design, data collection and analysis, decision to publish, or preparation of the manuscript.

**Competing interests:** The authors have declared that no competing interests exist.

between the landscapes, *T. cruzi*, arbovirus, and *Leptospira* infections were the most prevalent in urban and rural settlements, whereas *Leishmania* and *Rickettsia* had similar prevalence across habitats, likely due to the diversity and abundance of the infected vectors involved. The prevalence of *T. gondii* and *Leptospira* spp. may reflect poor hygiene conditions. Additionally, results suggest that prevalence of zoonotic and vector-borne diseases is higher in deforested areas and agricultural aggregates, and in sites with precarious health and infrastructure services.

## Conclusions

Some hosts, vectors, and transmission trends of zoonotic and vector-borne diseases in the YP are well known but others remain poorly recognized. It is imperative to reinforce practices aimed at increasing the knowledge, monitoring, prevention, and control of these diseases at the regional level. We also emphasize the need to perform studies on a larger spatio-temporal scale under the socio-ecosystem perspective, to better elucidate the interactions between pathogens, hosts, vectors, environment, and sociocultural and economic aspects in this and many other tropical regions.

### Author summary

Modification of the landscape and ecosystems is tightly linked with the transmission of zoonotic and vector-biorne diseases, with crucial consequences on public health. We reviewed the published literature (2015–2024) on the most frequent vectors, reservoirs and hosts, the prevalent infections, as well as the factors associated with transmission risk and the anthropogenic landscape modification in the Yucatan peninsula, Mexico. The most relevant pathogens of zoonotic risk included *Trypanosoma cruzi*, arboviruses, Leishmania, Rickettsia, Leptospira, and *Toxoplasma gondii*. Our summary suggests that prevalence is higher in deforested areas and agricultural aggregates, and in sites with precarious health and infrastructure services. Furthermore, the spatial distribution of zoonotic agents in the peninsula highlights the need to better understand the impact of land use changes on zoonoses.

## Introduction

The Yucatan peninsula (YP) is located along southeast Mexico and encompasses the states of Campeche, Yucatan, and Quintana Roo. This region has a tropical subhumid climate and possesses the second largest area of forest cover in Latin America, after the Amazon basin [1,2]. However, like other tropical regions, the YP has experienced significant social and economic changes that have impacted the use and appropriation of its natural resources [3,4]. Historically, development policies implemented since 1970 have led to a significant modification of the landscape with negative effects on forest cover, biological diversity, and ecosystem processes. Campeche, Yucatan, and Quintana Roo represent a general gradient of landscape modification, where the first two have higher deforestation trends, mainly for agriculture and cattle raising, while Quintana Roo harbors more extended forested areas, with habitat modification predominantly along the coast because of urbanization and tourism infrastructure [5]. The north of Yucatan experienced significant transformation since colonial times due to

'henequén' cultivars, while intensive urbanization has occurred in recent times. Additionally, activities like infrastructure construction (e.g. electrical system, communication and transportation roads), hydrological alteration, and various productive enterprises have impacted the entire YP. All these have generated gains for "human well-being" and economic development; however, at a high environmental cost on the ecological integrity of ecosystems, their structure, function, and resilience [2,6,7]. In the period between 2019–2023, the YP lost 285,580 ha of forested landscape, at a rate of 0.4% [8]. These patterns of environmental degradation are expected to continue and potentially increase in the forthcoming decades.

Habitat modification and land use changes not only impact ecological interactions but also alter the relationships between humans and nature. A clear example is the increase in the transmission of vector-borne (e.g. dengue, Zika, chikungunya) and zoonotic diseases. Vector-borne pathogens are transmitted mainly by arthropods, whereas zoonotic infections are naturally transmitted and shared between vertebrate animals and humans bidirectionally, including bacteria, viruses, macroparasites, and prions [9]. Both diseases emerge because of eco-evolutionary processes, where pathogens adapt to new niches and hosts [10]. The concurrent biodiversity loss and the increment of vector-borne and zoonotic diseases around the world support the suggestion that the two phenomena are intertwined [11,12].

Anthropogenic habitat modifications affect the transmission of vector-borne and zoonotic diseases by modifying the composition, abundance, distribution, and behavior of vectors (organisms that transmit diseases) and susceptible hosts [7,13,14]. These modifications create human-animal interfaces that facilitate interactions among humans, domestic animals, and wildlife, including their associated microbiomes [6]. These microbiomes may include pathogens such as bacteria (e.g. *Salmonella* spp., *Bartonella* spp., *Brucella* spp., *Borrelia* spp., *Leptospira* spp.), parasites (e.g. *Plasmodium* spp., *Leishmania* spp., *Trypanosoma cruzi*), and viruses (e.g. dengue, Zika, rabies, West Nile virus). Notably, these pathogens can be found in the YP and pose significant risks to human health [15–19].

Nearly 60% of human infectious diseases arise from endemic and enzootic zoonoses–both refer to diseases that naturally occur in animals and can be transmitted to humans [10]. Although zoonotic diseases have been investigated in the YP since the 1960s, they have not been successfully controlled and, in fact, an increase in their incidence has been observed [1]. Considering that a high zoonotic risk has been pointed out in tropical regions associated with changes in land use, our objective was to evaluate this premise for the YP. To this end, we conducted a review of bibliographic sources covering the most recent decade on vector-borne and zoonotic diseases in the region, aiming to: 1) integrate and synthesize current available information on vectors, reservoirs, and hosts of vector-borne and zoonotic pathogens; 2) analyze aspects of prevalence, transmission, and the most significant infections; and 3) identify potential factors associated with transmission risk and with anthropogenic landscape modification.

## Methods

We conducted a systematic review on vector-borne and zoonotic pathogens in the Yucatan peninsula (YP), considering the PRISMA criteria to identify, collect, select, and analyze studies and their relevant information [20]. The search encompassed from 2015 to 2024 and covered recently published sources, thus this work does not present the historical context on the topic, which has been reviewed elsewhere [see 1, 19]. We searched different databases including SciELO [21] and LILACS [22] to access the main sources of scientific and technical literature from Latin America and the Caribbean [23], as well as ScienceDirect [24], NCBI PubMed [25], and Google Scholar [26].

The search terms included "wildlife microbiome", "zoonotic diseases", "zoonosis", "Trypanosoma cruzi", "Leishmania", "Toxoplasma gondii", "Leptospira", "Rickettsia", "Dengue virus", "Zika virus", "chikungunya virus", "arboviruses", and "vector-borne diseases", combined with "Campeche", "Quintana Roo", and "Yucatan". Research articles that met the search criteria were selected, as well as book chapters and other literature reviews with content verification to cover other information sources.

From the selected material, we extracted all relevant information and saved it in a database available as Supporting information (S1 Table). The information recorded in this database includes year and type of publication, name of the journal, social sectors considered in the study (e.g. general population, government, and health and research sectors); date, location (geographical coordinates if available), and site where the sampling was conducted. Site refers to human settlements in rural and urban areas (domestic and peri-domestic habitats) as those most disturbed, ecotones where clearings for grazing and agriculture predominated, and the conserved areas (sylvatic) included forests with patches of native vegetation of varying extent. Other relevant data was also included when available, such as land use, pathogen, vectors and hosts or reservoirs identified, sample size, prevalence, and diagnostic method (S1 Table).

The variables that were considered for the analysis in this review were the following: pathogen, collection site (rural, urban, ecotone, and sylvatic), region in the YP (Campeche, Quintana Roo, Yucatan), seroprevalence (SP), number of organisms (Ni) collected per species, and diagnostic technique. Given the variation in Ni, we weighted the prevalence (Pw) by multiplying the log10-transformed Ni (to achieve a linear distribution) x SP [27,28]. This data subset was used to evaluate the dynamics of zoonotic pathogens transmission through the detection of infected vectors, reservoirs, and hosts in sites with different levels of anthropogenic modification, considering also the Pw. In addition, information on the number of positive cases in humans of potential zoonotic diseases was obtained from the Bulletin from the Mexican Epidemiological Surveillance System [29], considering the accumulated cases recorded to week 52 per year from 2015 to 2024.

An analysis of variance (ANOVA) was used to determine statistical differences (P<0.05) of Pw between collection sites with different levels of anthropogenic modification. Urban sites represented the most modified habitat, followed by rural, ecotone, and sylvatic. The zoonotic and vector-borne pathogens and the Pw data in the YP from 2015–2024 were recorded on a map of geostatistical boundaries [30] and land use [31], respectively, using the georeferencing information provided in each study. The approximate geolocation of the localities reported without coordinates was obtained using Google Earth [32]. The maps were processed using ArcGIS 10.8 software.

## Results

### Bibliographic sources

A total of 165 bibliographic sources published between 2015 and 2024 related to vector-borne and zoonotic pathogens in the Yucatan peninsula (YP) were evaluated. The number of publications varied between five and 24 studies per year in the period reviewed, without increasing in the last years (Fig 1). The highest number of studies corresponded to Yucatan (66.7%, 110/165), followed by Campeche (13.9%, 23/165) and Quintana Roo (7.3%, 12/165). Other publications (12.1%, 20/165) covered more than one state in the YP or did not specify the study location within the region.

From the evaluated publications, 116 (70.3%) were research articles, 15 (9.7%) were short communications including notes and scientific communications, 20 (12.1%) were case reports related to clinical presentation, diagnosis, and/or treatment, eight (4.8%) provided literature

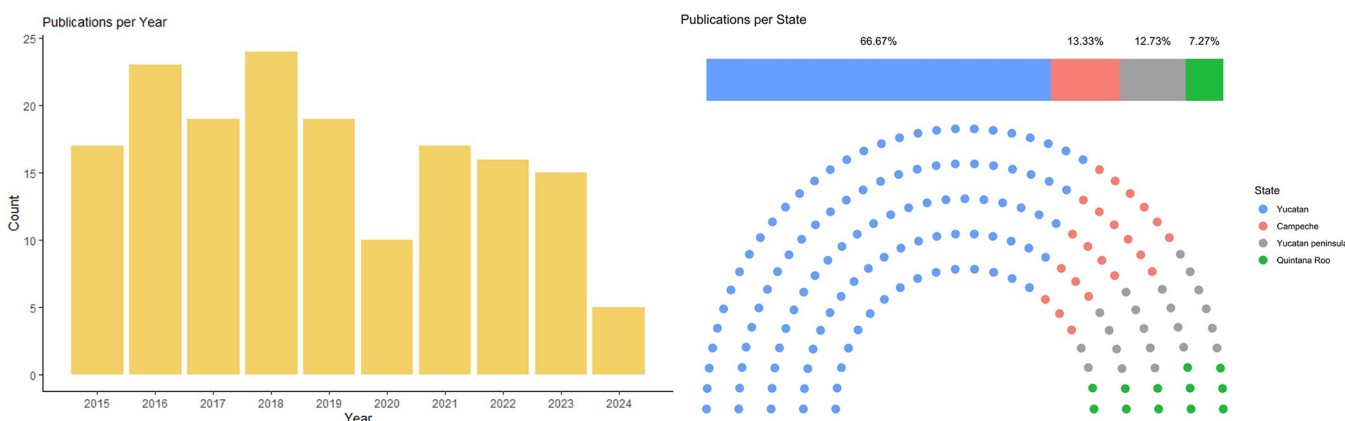

**Fig 1. Number of publications per year and state in the Yucatan peninsula (YP).** Number of publications per year (left) and by state (right) that reported zoonotic and vector-borne diseases in the YP during the period of 2015 to 2024.

reviews, and four (2.4%) were book chapters (Fig 2). Additionally, two editorial notes (0.6%) were found. None of the reviewed publications addressed the study of the microbiome of wildlife.

The studies were published in 88 specialized journals in zoonotic diseases, public health, veterinary medicine, virology, parasitology, tropical medicine, ecological health, evolution, and genetics of infectious diseases, wildlife diseases, entomology, and tropical diseases. The journals with the highest number of publications under the search criteria were Acta Tropica (12), PLoS Neglected Tropical Diseases (8), Revista do Instituto de Medicina Tropical de São Paulo (7), Southwestern Entomologist, and Zoonoses and Public Health (both with 5 publications).

The research groups performing the studies belong to national universities, with the Universidad Autónoma de Yucatán (UADY) and the Centro de Investigaciones Regionales "Dr. Hideyo Noguchi" with the highest number. Collaborative studies involved national public health institutions and vector control centers such as Instituto Mexicano del Seguro Social (IMSS), Instituto Nacional de Pediatría (INP), Instituto de Diagnóstico y Referencia Epidemiológicos (InDRE), Secretaría de Salud, Departamento de Control de Vectores del Estado de Yucatán, Dirección de Prevención y Protección de la Salud de los Servicios de Salud del Estado de Yucatán, and multiple regional hospitals, among others. Various sectors participated, including the Federation, local government, zoos, civil associations (e.g. Mexican Association of Conservation Medicine, Kalaan-Kab AC), local leaders, general population, and industry (e.g. pig farming industry). Some of the analyzed studies were conducted in collaboration with

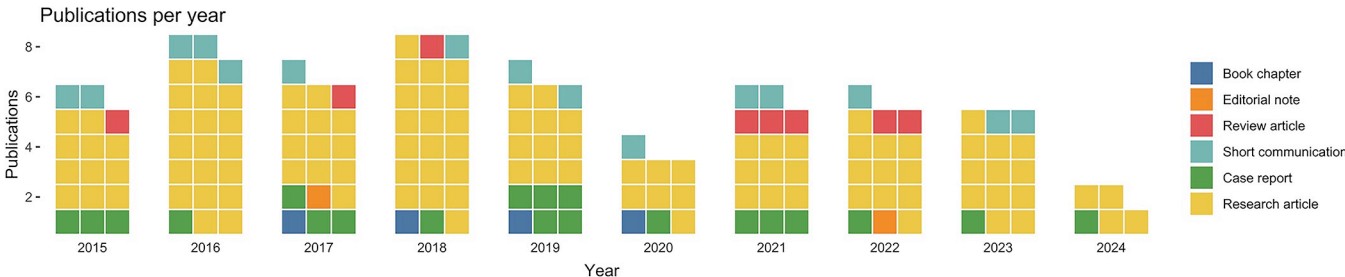

**Fig 2. Type of publications.** Scientific studies addressing vector-borne and zoonotic diseases in the Yucatan peninsula per year.

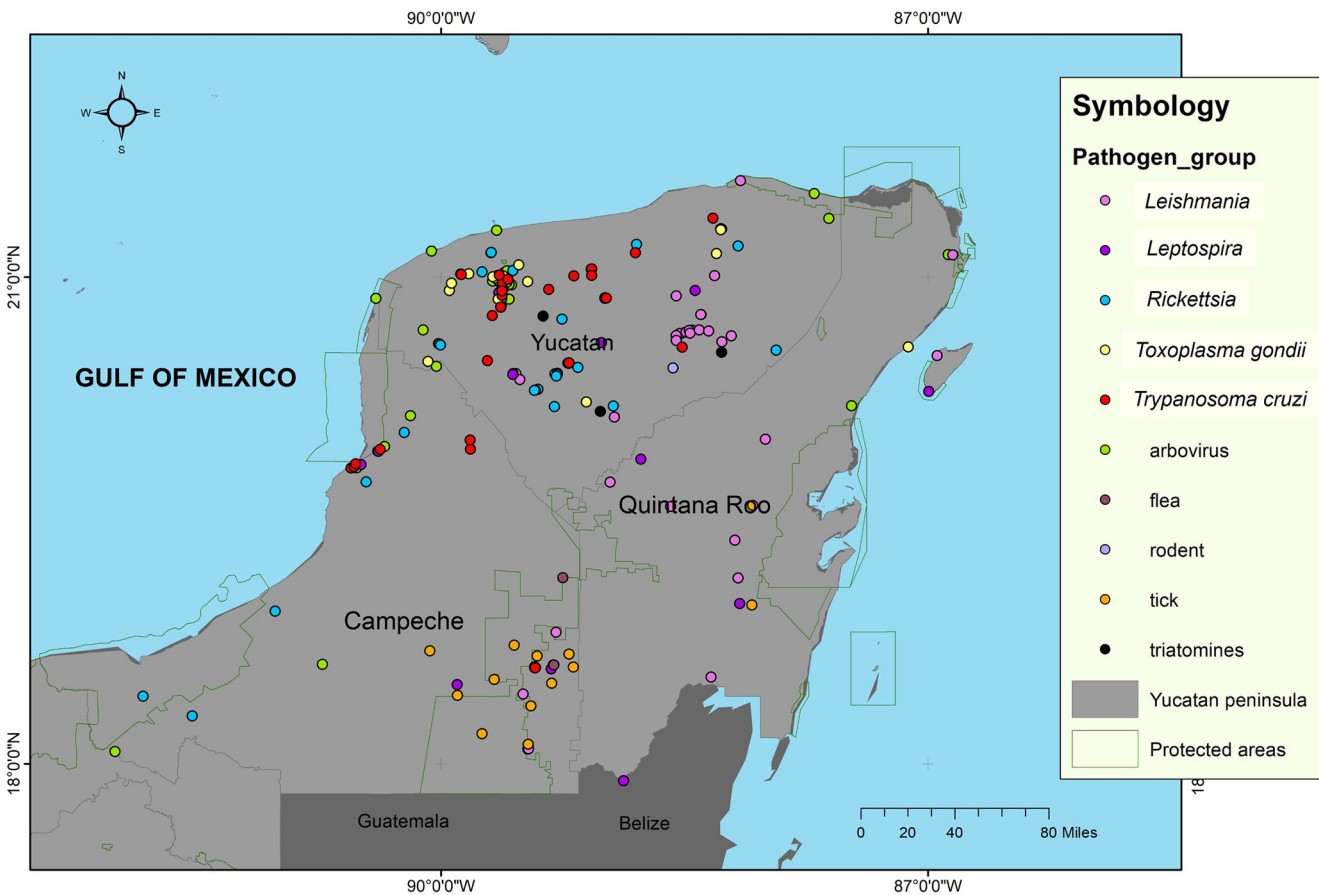

**Fig 3. Map of sampling sites.** Sampling sites for the study of zoonotic and vector-borne diseases during the evaluated period (2015 to 2024), indicating as well sites of some vectors and reservoirs. Map of geostatistical boundaries from public domain [30; https://www.gits.igg.unam.mx/idea/descarga] and map created using ArcGIS 10.8 software. The states of Yucatan, Campeche, and Quintana Roo of the Yucatan peninsula are depicted.

foreign universities and research centers from countries including Argentina, Brazil, Colombia, Cuba, Guatemala, Germany, Italy, United States of America, France, and the United Kingdom.

The studies in Yucatan were conducted in 31 localities located in 61% of the municipalities (65/106), in Campeche in 31 localities distributed in 77% of the municipalities (10/13) and in Quintana Roo in 82% of the municipalities (9/11). However, considering the distribution of the sampling sites, a larger area was studied in Yucatan compared to Campeche and Quintana Roo (Fig 3). Most of the studies in Campeche were carried out in the Calakmul Biosphere Reserve and in Quintana Roo they were in the northeast and southeast of the territory.

## Study of zoonotic and vector-borne pathogens in the Yucatan peninsula

Based on this review, we identified that multiple factors motivated research on the dynamics of transmission of vector-borne and zoonotic diseases. The most frequent were the presence of vector-borne and zoonotic diseases in human and non-human mammals, high prevalences in human populations, close contact between humans, reservoir host animals and vectors, production of meat for human consumption, water quality, environmental contamination, and distribution and abundance of potential reservoirs and vectors in different types of habitats (urban, rural, ecotone, and sylvatic). Other factors included accessibility and proximity to

research centers, monitoring of prevention and control programs, as well as the willingness of the people to share information and to participate in sample collection and implementation of vector-borne and zoonotic disease control programs.

The studies included parasites (i.e. nematodes, helminths, cestodes, protozoa), viruses, and bacteria. Most studies focused on *Trypanosoma cruzi* (Chagas 1909) (36 studies), arboviruses, and bacteria from the genus *Rickettsia* (da Rocha-Lima 1916) (29 studies each), *Leishmania* (Ross 1903) (21 studies), *Leptospira* (Johnson & Staton 1964) (14 studies), and *Toxoplasma gondii* (Nicolle & Manceaux 1909) (11 studies) (Fig 3). Four of these pathogens, i.e. *T. cruzi*, arboviruses, *Rickettsia*, and *Leishmania*, are primarily transmitted from hematophagous arthropods to humans and other vertebrate reservoir hosts. *Leptospira* and *T. gondii* are transmitted through contact of mucous membranes and wounds with infected materials, and via ingestion of contaminated food and water.

Other detected parasites transmitted by arthropods were *Babesia* spp. (Starcovici 1893), *Dipylidium caninum* (Linnaeus 1758), *Dirofilaria immitis* (Leidy 1856), *Hymenolepis diminuta* (Rudolphi 1819), and *H. nana* (Ransom 1901). Among the mosquito-borne viruses, various flaviviruses were identified, including St. Louis encephalitis virus (SLEV5, strain TBH-28), West Nile virus (WNV, strain NY99-35261-11), dengue virus (DENV-2, strain NGC and DENV-4, strain 241), and Houston mesonivirus (HOUV). Additionally, the lyssavirus-causing rabies (RABV) and two vertebrate-specific flaviviruses identified as modoc (modoc in Williams 2015; MODV, strain M544) and Apoi virus (APOIV) were also reported. Mosquito-specific viruses with unknown zoonotic potential included *Culex flavivirus* (CxFV-T1123), Mayapan nodavirus (MYPV H44 and H56), Merida rhabdovirus (MERDV), and Uxmal negevirus (UXMV M985, M1000, and M2038). Among the bacterial vector-borne pathogens, *Bartonella* spp., *Borrelia burgdorferi*, *Ehrlichia ewingii*, and *E. canis* were also found.

Pathogens were detected with up to five standardized protocols, where molecular techniques (e.g. PCR, quantitative PCR (qPCR)), restriction fragment length polymorphism (RFLP), and sequencing were the most frequent. Immunological tests included enzyme-linked immunosorbent assay (ELISA), indirect immunofluorescence (IIF), plaque reduction neutralization test (PRNT), antibody fluorescence (FAT), and microscopic agglutination tests (MAT). Parasites were also detected via microscopy (Table 1).

According to the Bulletin from the Mexican Epidemiological Surveillance System [29], between 2015 and 2023, leishmaniasis, tripanosomiasis, and dengue had the highest number of reported cases in humans in the YP. Yucatan accounted for the majority of the tripanosomiasis, dengue, rickettsiosis, and leptospirosis reports, while Quintana Roo presented the highest incidence of leishmaniasis and toxoplasmosis. In 2023 there was an increase of leptospirosis, leishmaniasis, and dengue cases (Fig 4).

## Vectors, reservoirs, and hosts of the most studied zoonotic and vector-borne pathogens in the Yucatan peninsula

*Trypanosoma cruzi*, the causative agent of Chagas disease, was detected in humans, dogs, cats, farm pigs, horses, and sheeps distributed in human settlements and ecotones (Fig 5). A total of 12 bat species were infected with *T. cruzi*, including *Artibeus jamaicensis* (Lech 1821), *A. lituratus* (Olfers 1818), *Carollia brevicauda* (Schinz 1821), *Chiroderma villosum* (Peters 1860), *Dermanura phaeotis* (Miller 1902), *Glossophaga soricina* (Pallas 1766), *Myotis keaysi* (Allen 1914), *Pteronotus parnellii* (Gray 1843), *Rhogeessa aeneus* (Goodwin 1958), *Sturnira lilium* (Geoffroy 1810), *S. ludovici* (Anthony 1924), and *S. parvidens* (Goldman 1917). Infected rodents (six species) included *Heteromys gaumeri* (Allen & Chapman 1897), *Mus musculus* (Linnaeus 1758), *Ototylomys phyllotis* (Merriam 1901), *Peromyscus yucatanicus* (Allen & Chapman 1897),

**Table 1. Detection tests of zoonotic and vector-borne pathogens in the Yucatan peninsula.**

| Disease | Etiologic agent | Diagnostic test | Reference |
|---|---|---|---|
| Bartonelosis | *Bartonella* spp. (b) | Culturing and PCR | [33,34] |
| Lyme disease | *Borrelia burgdorferi* (b) | PCR | [35] |
| Leptospirosis | *Leptospira* (b) | MAT, immunofluorescent imprint method, PCR, and sequencing | [35–40] |
| Rickettsiosis, Murine typhus | *Rickettsia* spp. (b) | Indirect immunofluorescence assay (IFA), MAT, PCR, RFLP, sequencing, and phylogenetic analysis | [40–46] |
| Vector-borne parasites | Babesia spp. (p), *Hymenolepis diminuta and H. nana* (c), *Dirofilaria immitis* (n), *D. cannis* (c) | Coprological analysis, sedimentation technique, PCR, and Sanger sequencing | [47–50] |
| Leishmaniasis | *Leishmania mexicana* (p) | qPCR, examination of cutaneous lesions by smear, biopsy, isolation-culture, PCR, and sequencing | [51–55] |
| Toxoplasmosis | *Toxoplasma gondii* (p) | IgG antibodies via ELISA assay, Isolation, qPCR, PCR, RFLP, cloning, sequencing, and genotyping | [56–59] |
| Chagas | *Trypanosoma cruzi* (p) | ELISA, PCR, IIF technique; qPCR coupled with HRM analysis, western blot, cloning, sequencing, microscopy | [60–65] |
| Hematophagous arthropods ectoparasites | Mites, lice, mites, ticks, mosquitoes, sand flies (vectors) | Direct and molecular identification by PCR | [66–68] |
| Dengue, yellow fiber, chikungunya, Zika, Western Nile virus, Venezuelan equine encephalitis | DENV, YFV, CHIKV, ZIKV, WNV, VEEV (v) | Cell culture, virus isolation, ELISA, immunofluorescence of brain tissue, PCR, RT-PCR, Sanger sequencing, and massive parallel sequencing | [69–72] |
| Rabies | RABV (v) | FAT, RT-PCR, and sequencing | [73] |

Bacteria (**b**), cestode (**c**), nematode (**n**), protozoan (**p**), virus (**v**).

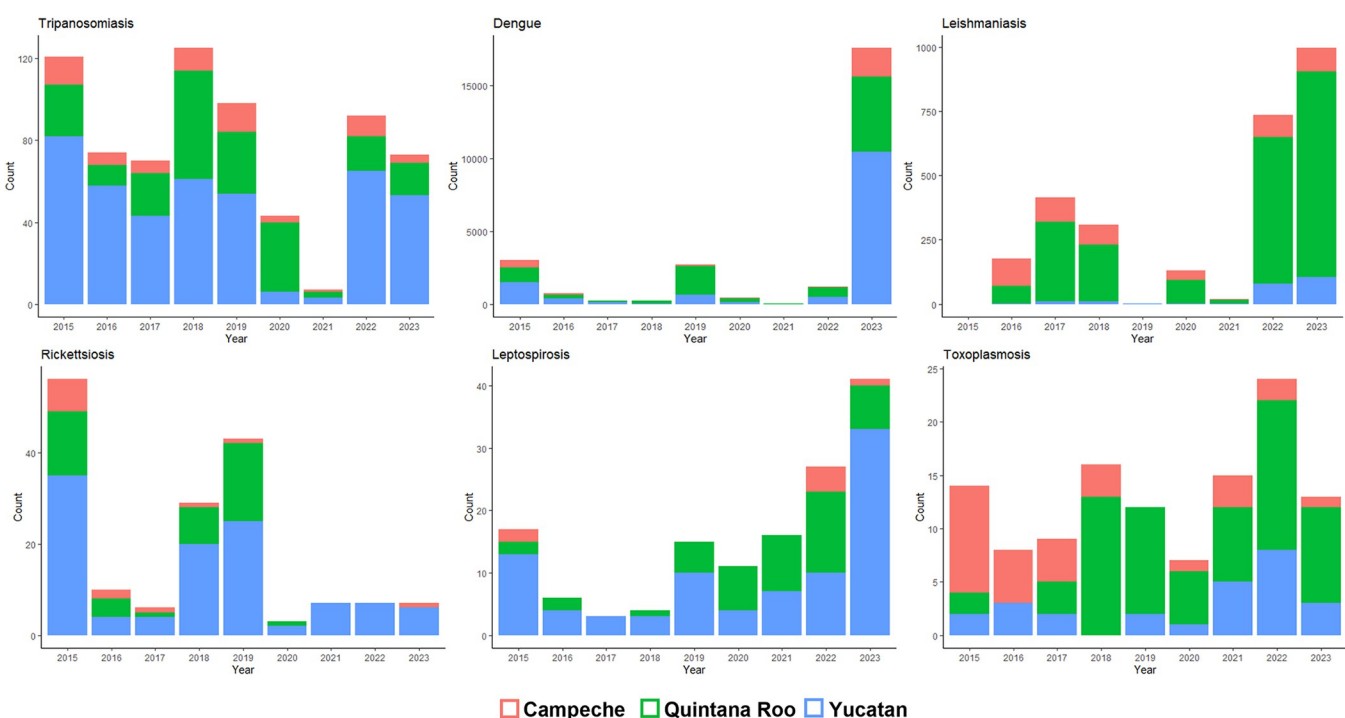

**Fig 4. Number of human cases of likely zoonotically transmitted diseases.** Data obtained from the Bulletin from the Mexican National Epidemiological Surveillance System [29]. Counts correspond to the accumulated cases reported up to week 52 of each year from 2015 to 2023.

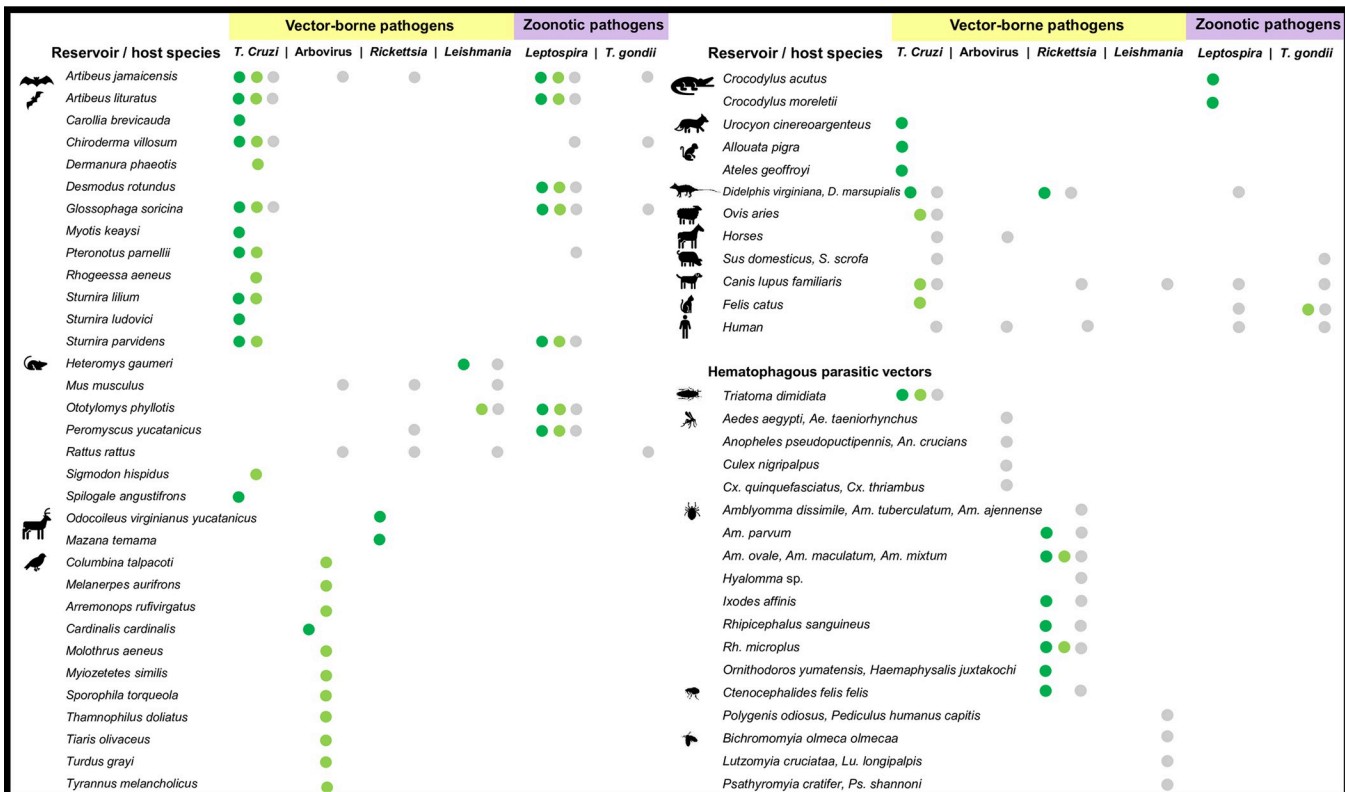

**Fig 5. The most studied zoonotic and vector-borne pathogens.** Infected species including vectors and reservoirs and/or hosts collected in habitats with different degrees of anthropogenic modification in the Yucatan peninsula, based on the reviewed studies from 2015 to 2024. Green: sylvatic, light green: ecotone, gray: urban. Animal silhouettes obtained from microsoft 365 software (account maria.sanchez-soto@cinvestav.mx) and from the open source www. phylopic.org, freely available for reuse under Creative Commons licenses.

*Rattus rattu*s (Linnaeus 1758), and *Sigmodon hispidus* (Say & Ord 1825). Other mammals infected with this parasite were non-human primates (*Alouatta pigra*, Laurence 1933, and *Ateles geoffroyi*, Kuhl 1820*)*, opossums (*Didelphis virginiana*, Kerr 1752), and spotted skunks (*Spilogale angustifrons*, Howell 1902).

Hematophagous arthropods such as *Triatoma dimidiata* and *T. nitida* (Latreille 1811) were the only reported triatomine (Hemiptera: Reduviidae) species (Fig 5). Infected *T. dimidiata* bugs, the main vector of *T. cruzi*, showed a significantly higher abundance in rural and sylvatic sites (ANOVA, $P < 0.05$). Higher numbers of infected triatomines were reported within houses and peak abundances in warm and dry seasons regardless of the habitat.

Mosquito-borne viruses were studied in humans, pigs, bats, birds, and rodents (Fig 5). Molecular and serologic tests showed positive results for dengue (DENV), Zika (ZIKV), and chikungunya virus (CHIKV) in human populations. ZIKV was detected in *Sus scrofa* pigs (Linnaeus 1758) and *A. jamaicensis* bats; this flying mammal species tested positive for West Nile virus (WNV) infections. *Rattus rattus* and *M. musculus* rodents collected in rural areas tested positive for dengue (DENV2, DENV4), Modoc virus (MODV), St. Louis encephalitis virus (SLEV), Apoi virus (APOIV), yellow fever virus (YFV), and WNV. Other WNV infected organisms belonged to 11 bird species captured in ecotones and sylvatic sites. Female mosquitoes were identified as an important hematophagous vector in the transmission of arbovirus in human settlements. A total of 34 mosquito species from 13 genera (Diptera: Culicidae) were collected in rural and urban settlements, from which *Aedes aegypti* (Meigen 1818), *Ae.*

*taeniorhynchus* (Wiedemann 1821), *Anopheles crucians* (Meigen 1818), *An. pseudopuctipennis* (Theobald 1901), *Culex nigripalpus* (Theobald 1901), *Cx. quinquefasciatus* (Say 1823), and *Cx. thriambus* (Dyar 1921) resulted positive for the Alphavirus DENV, ZYKV, and CHIKV. Mosquitoes were found in higher numbers in the rainy season compared to the dry season.

Protozoan parasites of Leishmania (*Leishmania* spp., *L. americana*, and *L. infantum*) were detected in humans from rural environments with activities in sylvatic areas, as well as in urban dogs and rodents (*H. gaumeri* and *O. phyllotis*) in rural, ecotone, and sylvatic sites (Fig 5). Infected rural Phlebotomine sand flies (Diptera, Psychodidae, Phlebotominae) with *Leishmania* corresponded to the species *Psychodopygus panamensis* (Shannon 1926), *Bichromomyia olmeca olmeca* (Fairchild & Hertig 1957), *Lutzomyia cruciata* (Coquillet 1907), *Lu. longipalpis* (Lutz & Neiva 1912), *Psathyromyia cratifer* (Fairchild and Hertig 1961), and *Pa. shannoni* (Lane and Cerqueira 1946).

Bacteria of the genus *Rickettsia* were detected in human and non-human mammals in all the habitats (Fig 5). Positive tests were obtained from human populations (*R. parkeri*, *R. rickettsii*, *R. typhi*, *Orientia tsutsugamushi* belonging to the Rickettsiaceae family), dogs (*Rickettsia* spp., *R. typhi*, *R. parkeri*), *A. jamaicensis* bats, and *R. rattus*, *M. musculus*, and *P. yucatanicus* rodents (*R. parkeri* and other species) in rural areas. White-tailed deer (*Odocoileus virginianus yucatanensis*, Zimmermann 1780), Mazama deer (*Mazama temama*, Kerr 1792), and opossums from sylvatic and rural sites (*Didelphis virginiana*, Kerr 1792, and *D. marsupialis*, Linnaeus 1758) were positive for tick-borne rickettsial agents.

The arthropods transmitting *Rickettsia* species included fleas (Syphonaptera: Pulicidae) (*Polygenis odiousus*, Smit 1958, and *Ctenocephalides felis*, Bouché 1835) and 15 ticks: *Amblyomma cajennense* (Fabricius 1787), *A. dissimile* (Koch 1844), *A. tuberculatum* (Marx 1894), *A. maculatum* (Koch, 1844), *A. mixtum* (Koch 1844), *A. ovale* (Koch 1844), *A. parvum* (Aragão 1908), *Haemaphysalis juxtakochi* (Cooley 1946), *Hyalomma* sp. (Koch 1844), *Ixodes affinis* (Neumann 1899), *Ornithodoros yumatensis* (Koch 1837), *Rhipicephalus sanguineus* (Latreille 1806), and *Rh. microplus* (Canestrini 1888) collected from rodents, dogs, pigs, deers, and felids (Fig 5).

*Leptospira* infections were found in bats *A. jamaicensis*, *A. lituratus*, *Chiroderma villosum* (Peters 1860), *Desmodus rotundus* (Geoffroy 1810), *P. parnellii* (Gray 1843), and *Sturnira parvidens* (Goldman 1917), rodents (*H. gaumeri*, *O. phyllotis*, *P. yucatanicus*, and *R. rattus*), humans, and companion animals (dogs and cats) (Fig 5). *Leptospira* were also identified in wild crocodiles belonging to several serotypes detected by microscopic agglutination test (MAT). The identified serotypes included *autumnalis*, *bataviae*, *bratislava*, *canicola*, *grippotyphosa*, *hardjoprajitno*, *icterohaemorrhagiae*, *pomona*, *pyrogenes*, *tarassovi*, and *wolffi*.

Infections caused by the protozoan parasite *T. gondii* were found across habitats, with most reports detected in the rural sites. Positive reports for *T. gondii* infections included opossums (*D. virginiana*), black rats (*R. rattus*), domestic animals (pigs, cats, and dogs), bats, and humans (Fig 5). Types I, II, and III were identified in pigs and cats. Two studies reported the presence of *Leptospira* in water and playground sandboxes in urban sites.

## Prevalence in habitats with different anthropogenic modification level

Zoonotic and vector-borne pathogens were studied in sites with different degrees of anthropogenic modification and land cover. Most of the studies (56.48%, 9348/16,574) were conducted in rural sites, followed by urban, sylvatic, and ecotones, accounting for 24.9% (41/165), 18.8% (31/165), and 7.9% (13/165) of the total sources reviewed, respectively (Fig 6). Urban settlements represented the habitat with the highest anthropogenic modification, followed by rural areas usually consisting of houses built with stones, wooden posts, palm leaf roofs, and

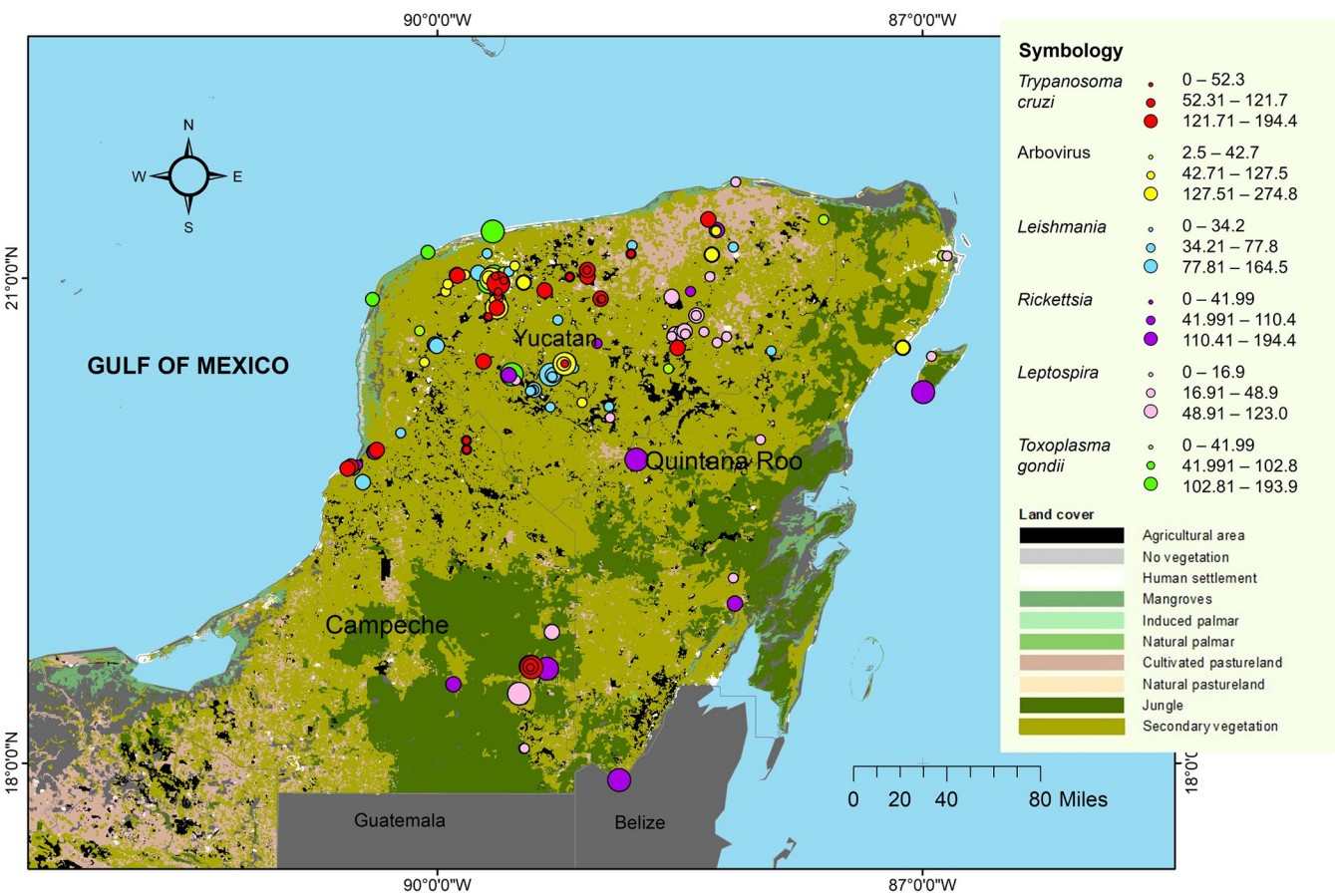

**Fig 6. Land cover and sampling sites for the study of zoonotic and vector-borne pathogens in the Yucatan peninsula.** Weighted prevalence (Pw) of pathogens reported in the period of 2015–2024. Land cover categories defined by public domain [31; https://earth.google.com/web/search/] and map created using ArcGIS 10.8 software. The states of Yucatan, Campeche, and Quintana Roo of the Yucatan peninsula are depicted.

concrete block interior walls, with large backyards harboring domestic animals (i.e. dogs, cats, poultry, pigs, horses, sheeps) and abundant native vegetation. Rural settlements were typically linked to the ecotone habitat where agricultural and livestock lands predominate. The sylvatic sites represented the most conserved habitat.

Overall, weighted prevalence (Pw) of vector-borne and zoonotic pathogens showed statistical variation (ANOVA, P<0.05) among collection sites (urban, rural, ecotone, and sylvatic) and pathogens (Table 2). Specifically, differences among collection sites were observed for *T. cruzi* and arboviruses, which had the highest prevalence in urban and rural settings. Other pathogens such as *Leptospira* spp. showed a higher incidence in sylvatic, rural, and urban sites. In contrast, no statistical differences were observed for *Leishmania* and *Rickettsia* species between sites.

**Table 2. Weighted prevalence differences considering zoonotic and vector-borne pathogens and collection sites (urban, rural, ecotone, and sylvatic) in the Yucatan peninsula.**

| Factor | Total sum of squares | Within-group sum of squares | F | P-value |
|---|---|---|---|---|
| Pathogen | 513518.91 | 49176.60 | 4.43 | <0.0008 |
| Collection site | 513518.91 | 40385.98 | 3.03 | <0.0074 |

Three pathogen groups were studied in all the habitats, *T. cruzi*, *Leptospira*, and *Leishmania* spp. Regarding host and collecting site, the highest *T. cruzi* prevalences occurred in dogs (mean±standard deviation, 71.32±78.65), opossums (63.30±66.79), pigs (56.64±73.35), humans (53.34±60.46), and triatomines *T. dimidiata* (67.62±32.11) from domestic and peridomestic environments. The highest *Leptospira* infections were found in wild crocodiles (88.85 ±24.08), dogs (58.6±43.77), bats (31.71±17.97), and rodents (18.18±12.42), while *Leishmania* mainly infected humans (45.33±58.16), rodents (37.47±29.74), and sand flies (22.74±26.27) present in rural settlements surrounded by local vegetation. The highest arbovirus incidence was detected in humans (93.17±89.02), mosquitoes (53.75±2676.87), and rodents (37.53 ±56.6), while for *Rickettsia* was in rodents (34.30±14.24), dogs (74.93±27.54), humans (25.85 ±31.76), and ticks (45.27±33.89) that fed on domestic and synanthropic animals (e.g. dogs, cats, horses, opossums, deers). Finally, dogs (192.10±2.55), cats (98.74±5.75), and pigs (109.45 ±58.65) distributed in rural sites had the highest prevalence of *T. gondii* (Table 3).

## Discussion

The presence, transmission, and prevalence of vector-borne and zoonotic pathogens varies depending on multiple factors related with climate, ecology, evolution, landscape, and socioeconomics, among others. Although generalizations can be made regarding the patterns of distribution of zoonotic and vector-borne pathogens, it is crucial to determine and understand the specific factors of particular regions that can significantly contribute to the surveillance, prevention, and control of zoonoses. Based on the information gathered from the studies included in this review, we highlight key aspects about zoonotic and vector-borne pathogens in the Yucatan peninsula (YP), as well as the factors that may influence their endemic status and emergence in the region.

### Zoonotic and vector-borne pathogens, transmission cycles, and prevalences

The most relevant zoonotic pathogens identified were *Trypanosoma cruzi*, arboviruses, *Leishmania* (*L. mexicana*, *L. braziliensis*, *L. infantum*), vector-borne *Rickettsia* species (*R. typhi*, *R. parkeri*, *R. rickettsii*, *O. tsutsugamushi*, *R. endosimbiont*, *R. lusitane*, *R. felis*), *Leptospira* (e.g. *L. interrogans*, *L. infantum*, *L. noguchii*, *L. borgpetersenii*, *L. santarosai*), and T. *gondii*. However, there are other important infections in terms of public health for the region (Table 1).

Pathogens of zoonotic risk can be divided into three main groups: those that are transmitted by direct contact with animals (e.g. rabies virus); those transmitted indirectly through contact of mucous membranes and wounds with infected materials and ingestion of contaminated food and water (*Leptospira* spp., *T. gondii*, *Brucella* spp.); and those transmitted by vectors including arthropods (fleas, lice, ticks, mosquitoes) that can transmit bacteria, parasites, or viruses when they bite a host; for instance, West Nile Virus (transmitted by mosquitoes), Bubonic plague (transmitted by fleas), and *Rickettsia* (transmitted by ticks) [1,74]. The transmission of *T. cruzi* can also be oral, congenital, and can occur through blood transfusion and organ transplants [75,76]. It is important to emphasize that, regardless of their classification, these are causative agents of neglected infectious diseases [19,49,77–80]. Their high global prevalence disproportionately affects the poorest human communities, as their epidemiology is complex and often related to climatic characteristics, socioeconomic level, environmental degradation, precarious sanitary infrastructure, and limited access to quality healthcare services [81].

This analysis of the pathogen transmission cycle was mainly based on the identification of zoonotic and vector-borne pathogens in arthropod vectors and vertebrate reservoirs or hosts in different collecting sites (habitats). The results showed infected vectors and reservoirs/hosts

**Table 3. Highest weighted prevalences.** Weighted prevalences (Pw) from infected species collected in sites with different levels of anthropogenic modification (urban, rural, ecotone, and sylvatic) in the Yucatan peninsula.

| Pathogen | Highest prevalences | | | |
|---|---|---|---|---|
| | Infected species | Pw (Ni) | Site | State |
| *Trypanosoma cruzi | *Canis lupus familiaris* | 194.4 (88) | urban | Yucatan |
| | Human | 135.5 (81)/123 (17) | rural/urban | Yucatan |
| | *Triatoma dimidiata* (vector) | 128 (269) | rural | Yucatan |
| | *Didelphis virginiana* | 112.6 (35) | rural | Yucatan |
| | *Sus domesticus* | 108.5 (28) | rural | Yucatan |
| ◇Arbovirus | *Aedes aegypti* (vector) | 259.1 (2161) | urban | Yucatan |
| | Human | 152.8 (352), 237.1 (2978) | urban, rural | Yucatan |
| | *Rattus rattus* | 120 (75) | urban | Yucatan |
| | *Artibeus jamaicensis* | 42.7 (22) | urban | Yucatan |
| *Rickettsia* spp. | *Mus musculus* | 42.33–51.46 (13–32) | rural | Yucatan |
| | *Amblyomma spp.* (vector) | 71.98 (12) | rural | Yucatan |
| | Human | 74.8–72.5 (130–128) | rural | Yucatan |
| | *Canis lupus familiaris* | 100 (10), 97.52 (50) | rural | Yucatan |
| | *Rhipicephalus sanguineus* (vector) | 126.4 (380) | rural | Yucatan |
| *Leishmania* spp. | *Canis lupus familiaris* | 29.3 (61) | urban | Yucatan |
| | *Bichromomyia olmeca olmeca* (vector) | 49.2 (102) | rural | Campeche |
| | Human | 102.8 (13), 164 (146) | rural | Yucatan, Campeche |
| | *Pa. cratifer* (vector) | 72.1 (278) | rural | Yucatan |
| | *Lu. cruciata* (vector) | 74.8 (523) | rural | Yucatan |
| | *Ototylomys phyllotis* | 41.94 (5) | ecotone | Yucatan |
| | *Heteromys gaumeri* | 77.82 (6) | sylvatic | Yucatan |
| ◇*Leptospira* spp. | *Crocodylus acutus* | 123 (17) | sylvatic | Quintana Roo |
| | *Canis lupus familiaris* | 89.6 (96) | rural | Yucatan |
| | *Sturnira parvidens* | 67.7 (16) | all habitats | Yucatan |
| | *Artibeus jamaicensis* | 48.9 (22) | rural | Campeche |
| | *Peromyscus yucatanicus* | 39.4 (30) | all habitats | Yucatan |
| ◇*Toxoplasma gondii* | *Felis catus* | 102.8 (13), 94.7 (11) | rural, ecotone | Yucatan |
| | *Sus scrofa* | 171.8 (60) | rural | Yucatan |
| | Human | 176.6 (89) | rural | Yucatan |
| | *Canis lupus familiaris* | 193.9 (96) | rural | Yucatan |

* Pw with statistical differences between habitats (P <0.05).

◇Pw with statistical differences between diagnostic techniques (P <0.05).

Number of organisms collected (Ni).

with sylvatic, synanthropic, and domestic habits, suggesting that the distribution of zoonotic and vector-borne pathogens in the YP includes environments with different types of land coverage and degrees of anthropogenic modification (Fig 6). The highest weighted prevalences (Pw) were obtained from vectors, dogs, cats, pigs, humans, and some synanthropic rodents and bats in human settlements (Table 2). This highlights the importance of domestic animals and their hematophagous ectoparasites in maintaining infectious cycles in human settlements, while synanthropic animals (e.g. vectors, bats, rodents, marsupials) can connect transmission cycles between sylvatic and urban habitats [19,60]. Although infection prevalence aims to understand the transmission cycles of etiologic agents [80], such data depend on various factors and thus should be interpreted with caution. Differences in the prevalence may be biased

by the effort and sampling design or the sensitivity of the pathogen detection techniques (e.g. molecular and/or serological). They may also be related to host and vector population density, the ability of animals to adapt to anthropic environments, evolutionary processes of species, and even the implementation of vector and host control programs in different study sites, among others [40,48,51,60,82,83].

## Habitat conservation and the transmission of zoonotic and vector-borne pathogens

*Trypanosoma cruzi* and arboviruses showed significantly higher incidences in urban and rural settlements (Table 3). It is possible that landscape modification due to human intervention has affected the composition and abundance of vectors and hosts involved in the transmission cycles of some zoonotic pathogens [7,84]. Therefore, it is crucial to understand that zoonotic incidence is associated with different degrees of landscape conservation. While some species are negatively affected, others such as synanthropic rodents, bats, opossums, and arthropods may benefit from the habitat modification, adapting to live in deforested areas and maintaining a close relationship with human settlements [53,85], as those identified in this review (Figs 5 and 6). Deforestation has been related with the increase in the transmission of certain pathogens in human populations, including trypanosomiasis, leishmaniasis, arboviruses, and other pathogens in the region [53,85–87].

The degree of habitat conservation (e.g. vegetation cover, type of vegetation, soil and water pollution) also influences ecological processes and the associated ecosystem services (e.g. control of zoonotic pathogens). Although the most conserved habitats in general harbor greater diversity and frequency of potential zoonotic pathogens, vectors, and hosts in comparison with modified habitats, it is precisely the diversity of ecological communities in preserved environments that reduces the spread of such pathogens through the regulation of susceptible host populations [11,88]. This ecosystem service is known as the 'dilution effect of pathogens' [11,89]. The generality of the dilution effect hypothesis remains controversial, because multiple mechanisms can operate together in the same disease system (e.g. host quality, host and vector abundance), which challenges the empirical assessment of such systems [90]. On the other hand, metadata analyses have shown multiple evidence of dilution, independently of host density, study design, and type and specialization of pathogens [11]; also, some taxonomic groups that are likely sources of zoonotic pathogens tend to thrive when biodiversity is lost [91]. For instance, rural areas host various susceptible hosts that, due to their proximity to and/or productive activities performed in the surrounding forests, have greater contact with the conserved habitat, including vectors and natural hosts of known and potential zoonotic pathogens [51].

## Other factors related to zoonotic transmission in the Yucatan peninsula

It is noteworthy that over half of the studies considered in this review were conducted in rural areas (S1 Table), where diseases of animal origin are more frequently related to the combination of factors associated directly and indirectly with the interaction among pathogens, arthropod vectors, reservoirs, hosts, and the environment [92]. The density of domestic animals (i.e. dogs, cats, poultry, pigs, horses, sheep), larger vegetation coverage, and richness of synanthropic species in rural compared to urban areas, increases the interaction between people, domestic animals and wildlife, thus facilitating the transmission and dispersion of zoonotic pathogens [14].

The studies analyzed in this review frequently report rural sites (S1 Table) characterized by the construction of fragile houses, precarious sanitary conditions, and the absence of basic

services for water supply and waste management [44,93,94]. These factors are linked to direct and indirect exposure to zoonotic pathogens transmitted by hematophagous vectors and reservoir hosts [40]. Subsistence activities such as backyard animal breeding, wildlife hunting, and agriculture help meet food demands; however, they can amplify the transmission of zoonotic pathogens in the YP [51,95,96].

The intrinsic attributes of vectors, hosts, and pathogens, which determine the success of spread, replication, and transmission in animal and human populations [73,89], are also among the factors that act simultaneously and influence the dynamics of pathogen propagation [10]. We found that *T. cruzi* infected a higher number of vertebrate species in all habitats, while leishmania and arboviruses have impacted a significant number of people (Figs 4 and 5). These findings were based on published research and data from the Mexican Epidemiological Surveillance System, emphasizing the need for synergy collaboration among academia, the health sector, and government to establish effective strategies for surveillance, prevention, and control of zoonotic risk diseases.

Infected arthropod vectors (triatomines, mosquitoes, flies, ticks) exhibit that hematophagous parasites may play a central role in the transmission of various zoonotic pathogens (Fig 5) from infected to uninfected hosts. As ectothermic organisms, they thrive in warmer temperatures [97]. Indeed, increases in temperature and the intensity/duration of the dry season are ecological factors that have influenced vector specialization in feeding on human blood [98]. This, coupled with the growth of human populations, especially in urban areas of the region, increases the risk of vector-borne infectious diseases in humans in the YP, but also on a global scale [99].

When discussing zoonotic risks, international mobility and trade must be considered as they facilitate the circulation of potentially infected humans, animals, and products [92,100,101]. This increases the likelihood of spreading pathogens capable of infecting wildlife, domestic animals, and resident humans of the YP, a risk magnified by urbanization, globalization, tourism, and other human activities [10,92]. Indeed, humans can also be vectors that transport zoonoses from their natural enzootic systems, capable of establishing new reservoirs in other geographic areas [92,100]. Under this scenario, traditional strategies may be less effective in controlling imported zoonotic pathogens. This represents another crucial factor for the development of epidemics in the region, especially in urban areas [100], which has not been evaluated in the YP.

## Multifactorial causes of zoonotic disease transmission in the Yucatan peninsula

In the YP, the most common zoonotic diseases are those transmitted by vectors and include dengue, leishmaniasis, and trypanosomiasis [102] (Fig 4). These diseases are considered neglected according to the Pan American Health Organization [103] and represent a challenge for public health in the region. Furthermore, infections by pathogenic *Rickettsia* are also of concern due to their potential to affect different vectors and animal hosts and because of their impact on human health [40,45,46,104].

In response to the risks of zoonotic pathogen transmission, sustainable measures are required to control vector and host populations. This can include larval control through predators and bacteria, release of sterile vectors, elimination of possible breeding sites, proper waste management, conservation of natural habitats, and sterilization of companion animals [89,105,106]. For all these strategies to work in the long term, it is necessary to establish environmental education programs for One Health and to work together with local communities.

Evidence has shown that habitat modification by deforestation, urban expansion, and tourism, along with the effects of climate change, have an impact on the emergence and spread of

zoonotic diseases [98,107], which is a current scenario in the YP. It is crucial to update and strengthen intervention, control, mitigation, and adaptation strategies to these environmental modifications, adopting we insist a One Health approach [108], especially those leading to the protection of the structure and function of natural ecosystems. Urbanization plays a central role in the spread of zoonotic diseases. It is estimated that by 2050, 2/3 of the world population will live in urban areas [109], which makes it necessary to protect urban residents from animal-transmitted diseases. In the YP, the dengue is one of the most emerging diseases (Fig 4), but the circulation of other arboviruses-borne diseases, such as Zika, chikungunya, malaria, and West Nile virus, has also been confirmed, as well as viruses associated with bats, rodents, and other mammals [70,73,110–112].

The emergence of viral diseases has been linked to drastic changes in habitat and environmental conditions, both natural and caused by human activities [99]. These conditions can facilitate the movement of pathogen hosts to areas inhabited by humans in search of shelter and food; areas that also lack their predators (i.e. biological controls) [87]. In fact, arbovirus infections in the region are of great concern within the range of emerging diseases, so it is crucial to maintain monitoring programs to understand the patterns of viral diversity in wildlife and the factors influencing the successful transmission between different species [87].

In addition, population growth, urbanization patterns, expansion of agricultural land, international trade, and mobility of people are closely linked to the emergence of zoonotic diseases [7,84,104]. The indiscriminate use of antibiotics and insecticides also contributes to the persistence of zoonotic pathogens, as they may lose their effectiveness over time [106]. Finally, the low investment in disease surveillance and care, limited healthcare services, inadequate waste management, absence of prioritization guidelines from the health sector, and limited knowledge among susceptible populations (due to deficient communication programs), exacerbate the problem [76,77].

## Conclusions

Global trends in biodiversity loss, as well as the emergence and transmission of zoonotic diseases, have never been as relevant as they are nowadays. The COVID-19 pandemic has led to a crucial warning of the impact of zoonotic diseases on health, economy, and global stability, highlighting the need to understand zoonotic processes and risk factors. The pandemic underlined the importance of interdisciplinary protocols among different actors and institutions, as well as the need for further cooperation between academia and the health sector. The existing information on zoonotic and vector-borne diseases in the YP is heterogeneous, namely some hosts, vectors, and transmission trends are well known but others are not. Therefore, it is imperative to reinforce practices aimed at increasing the knowledge, monitoring, prevention, and control of these diseases at the regional level. Safeguarding health, social, and economic security by building tools and programs that promote the preservation of the ecosystem's structure and processes on which we depend, fostering synergy among the government, research, industry, and health sectors, and with society, is an urgent challenge. The conservation of natural ecosystems that solely depends on economic interests will have immeasurable and irreversible costs, impacting the less privileged populations. It is crucially urgent to take measures to protect the health of ecosystems, including controlling deforestation and protecting aquifers, with decision-making and planning instruments based on scientific knowledge. Finally, we emphasize the need of studies on a larger spatio-temporal scale, to better elucidate the interactions between pathogens, hosts, vectors, environment, sociocultural, and economic aspects in this and other tropical regions.

## Supporting information

**S1 Table. Information recorded from the revision of 165 bibliographic sources on vector-borne and zoonotic pathogens in the Yucatan peninsula.** The following information is included: year and type of publication, name of the journal, social sectors considered in the study (e.g. general population, government, and health and research sectors); date, location (geographical coordinates if available), and site where the sampling was conducted; land use, pathogen, vectors and hosts or reservoirs identified, sample size, prevalence, and diagnostic method were also included when available.
(XLSX)

## Author Contributions

**Conceptualization:** Ma. Fernanda Sánchez-Soto, Osiris Gaona, Luisa I. Falcón, Ella Vázquez-Domínguez.

**Data curation:** Ma. Fernanda Sánchez-Soto, Osiris Gaona, Ana Laura Vigueras-Galván, Luisa I. Falcón.

**Formal analysis:** Ma. Fernanda Sánchez-Soto, Osiris Gaona, Ana Laura Vigueras-Galván, Luisa I. Falcón.

**Funding acquisition:** Gerardo Suzán, Luisa I. Falcón, Ella Vázquez-Domínguez.

**Writing – original draft:** Ma. Fernanda Sánchez-Soto, Ella Vázquez-Domínguez.

**Writing – review & editing:** Ma. Fernanda Sánchez-Soto, Osiris Gaona, Ana Laura Vigueras-Galván, Gerardo Suzán, Luisa I. Falcón, Ella Vázquez-Domínguez.

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
