## [Decision Letter · Decision Letter 0]

24 Feb 2024

Dear Dr. Vázquez-Domínguez,

Thank you very much for submitting your manuscript "Prevalence and transmission of the most relevant zoonotic pathogens in the Yucatan peninsula: a review" for consideration at PLOS Neglected Tropical Diseases. As with all papers reviewed by the journal, your manuscript was reviewed by members of the editorial board and by several independent reviewers. In light of the reviews (below this email), we would like to invite the resubmission of a significantly-revised version that takes into account the reviewers' comments. 

Reviewers found the topic of value but recommend significant restructuring, adding a multitude of studies that should have been considered, revising the search strategy, and excluding non-zoonotic pathogens. Your revision should include a point-by-point response to reviewer comments.

We cannot make any decision about publication until we have seen the revised manuscript and your response to the reviewers' comments. Your revised manuscript is also likely to be sent to reviewers for further evaluation.

Sincerely,

Angela Monica Ionica, Ph.D.

Academic Editor

Justin Remais

Section Editor

The topic is of value but needs significant restructuring, as it does not include a multitude of studies which should have been considered. This is due to the search strategy. On the other hand non-zoonotic pathogens should be excluded.

Reviewer's Responses to Questions

**Key Review Criteria Required for Acceptance?**

**Methods**

-Are the objectives of the study clearly articulated with a clear testable hypothesis stated?

-Is the study design appropriate to address the stated objectives?

-Is the population clearly described and appropriate for the hypothesis being tested?

-Is the sample size sufficient to ensure adequate power to address the hypothesis being tested?

-Were correct statistical analysis used to support conclusions?

-Are there concerns about ethical or regulatory requirements being met?

Reviewer #1: Authors have identified using the PRISMA criteria to select studies from 2015 to 2023 with the search terms which they describe which primarily involves the term "zoonosis" or "zoonotic" and the Yucatan. 

I am concerned with the overall selection of the studies used in this article and the methods of identifying which studies to evaluate. For example, a recent study: "Trypanosoma cruzi in Mexican Neotropical vectors and mammals: wildlife, livestock, pets, and human population" PMID: 38060864, is not included in the investigation but should be given what the authors are wanting to analyze. 

The problem lies in the use of "zoonosis" or "zoonotic" and in pure, this refers to the infectious diseases which are spread directly from non-human host to a human. Often that term is not used in the title of a research article if not assessing directly for zoonotic transmission. I can only imagine if the authors included vector-borne in the selection criteria how many more studies would have been selected and thus this analysis is missing likely many studies. Also, it may be a better idea to include in the search terms the specific pathogen and Yucatan and other regions of interest. For example, Trypanosoma cruzi and Yucatan, Leishmania and Yucatan...

Reviewer #2: Although the methodology presented accords with the article's objective, I have concerns about two aspects of it: 1. the control of sample bias resulting from the literature review. Although a weighted presence is provided, the amount of articles found still has an impact on it. It is recommended that the writers clarify this point better. 2. Since serological detection could affect results, is it possible that there will be more variety if only studies using molecular detection are taken into account when calculating prevalence?

The weighted prevalence calculation is supported by the authors citing Vigueras (2019); nonetheless, it is preferable to credit the original works where this calculation is proposed. You can review the following papers:

1.Garamszegi LZ, Møller AP. 2007 Prevalence of avian influenza and host ecology. Proc. Biol. Sci. R. Soc. 274, 2003–2012. (doi:10.1098/rspb.2007.0124)

2. Walther BA, Clayton DH, Cotgreave PC, Gregory RD, Price RD. 1995 Sampling effort and parasite species richness. Parasitol. Today 11, 306–310. (doi:10. 1016/0169-4758(95)80047-6)

Reviewer #3: After reviewing the manuscript entitled "Prevalence and transmission of the most relevant zoonotic pathogens in the Yucatan peninsula: a review," here are the main comments and observations.

First, the manuscript is well written, including a relevant analysis of Zoonosis in the Yucatan peninsula in the last decade. The topic is pertinent because the region is experiencing rapid economic and urban development, significant environmental changes, and emerging diseases.

The methodological approach behind the review is sound, with proper documentation of the available bibliography.

Second, a significant issue for the manuscript is that some treated diseases are not zoonotic—specifically Blastocystis, Dengue, SIKA, and Chikungunya.

Third, the authors must include the proper analysis of the reviewed information on the ectoparasitic arthropods as zoonotic agents (not only vectors).

Fourth, the authors should revise the discussion of the findings because many of the paragraphs need more references, and some sections are speculative and not supported by the evidence provided in the revised manuscripts.

Fifth, the conclusions must focus on the evidence-based findings extracted from the reviewed papers.

**Results**

-Does the analysis presented match the analysis plan?

-Are the results clearly and completely presented?

-Are the figures (Tables, Images) of sufficient quality for clarity?

Reviewer #1: Given the methods need attention I am not sure we have accurate results.

Reviewer #2: The reported results are clear and in accordance with the methodology that was provided. If there is any relationship between the cases in the bibliographic review and the cases reported in the epidemiology bulletin, it would be nice to know it.

The results are categorized by state (Campeche, Yucatan, and Quintana Roo), however the introduction doesn't go into more depth on how the states differ from one another. It is suggested to be more precise so that readers unfamiliar with the studied geographic area would understand the variations identified by state.

Reviewer #3: Great figures.

Figure 5. Please change the beetle silhouette for a bug and the domestic fly silhouette for a mosquito without a beak.

L277. "Relative high number". Please be accurate about quantities and comparisons.

L280-281. Please provide the incidence values and comparison between them as evidence of the sentence.

L295. Check the spelling of scientific names. Also, it should stand for homogeny in writing scientific names and abbreviations of the genera. A suggestion would be to write the full name the first time, including authority, and then use the abbreviation for the genera. Use the first two letters of the genera in those cases where a genera start with the same letter. Remember that some genera have their proper abbreviation nomenclature, like Phlebotominae (See Marcondes CB. A proposal of generic and subgeneric abbre- viations for phlebotomine sandflies (Diptera: Psychodidae: Phlebotominae) of the world. Entomol News 2007;118(4): 351–356; doi: 10.3157/0013-872X(2007)118[351:APOGAS] 2.0.CO;2 and Marcondes CB. On the utilization of abbreviations for phlebotomine sand flies (Diptera: Psychodidae: Phlebotominae). J Med Entomol 2019;56(1):1; doi: 10.1093/jme/ tjy197).

L311-312. Aedes taeniorhynchus and Ochlerotatus taeniorhynchus are the same. Ochlerotatus was degraded to a subgenus of Aedes. Please revise the nomenclatural updates.

L315. Revise the scientific names of the Leishmania parasites. Some names should be corrected.

L348-349. There are two "finally". Please revise.

L389. Revise the scientific names of the Leptospira. Some names should be corrected.

L414. Revise the use of the word "demonstrates." The tone in a review manuscript is important.

L444-448. Please add references to support the arguments.

L486-487. Please add references to support the argument.

L515-516. Please add the proper citation format.

L519-520. Revise the sentence for clearance.

L526. Do you mean companion animals?

L529-531. Revise the tone of the sentence because it is speculative. The review and the results do not support the arguments. 

L538. Please add references to support the argument.

L551-554. Revise the sentence. The argument is out of the context of the paragraph.

L556-557. Please add references to support the argument.

L560-563. Please revise the tone of the argument. If there were a lack of information, this review would not be possible. But there is no lack of information; more information and evidence of zoonotic diseases in the region are probably needed.

**Conclusions**

-Are the conclusions supported by the data presented?

-Are the limitations of analysis clearly described?

-Do the authors discuss how these data can be helpful to advance our understanding of the topic under study?

-Is public health relevance addressed?

Reviewer #1: In my opinion, I think it is hard to make conclusions with what is being presented in this article.

Reviewer #2: The conclusions are clear, and the work's weaknesses are highlighted, especially with regard to the publishing bias. The health implications and their relationship to epidemiological surveillance are highlighted.

Reviewer #3: L567-571. Please revise the paragraph. The arguments are out of the context of the evidence gathered in the reviewed bibliography. Arguments are speculative for a conclusion.

L572-575. Again. The arguments in the paragraph are speculative since any documented viruses fall in the risk category that the authors describe. All described pathogens are endemic, but some are expanding their prevalence. However, particularly, the viruses considered in this review are already widely distributed in tropical regions of the world and have an endemic transmission state.

L577. Please rephrase the paragraph. The use of the word "heterogeneus" is confusing.

L582-588. Please revise the paragraph. The arguments are speculative for the evidence from the reviewed bibliography, particularly for the review's conclusion.

**Editorial and Data Presentation Modifications?**

Reviewer #1: (No Response)

Reviewer #2: In lines 106 and 107 the authors give as examples of viruses Nipah and SARS among others and then state that these can be found in the geographical region studied. It is suggested to be careful and clarify this point.

In line 392 the authors state that zoonoses are grouped into two groups; direct and indirect transmission. It is suggested to revise the classification because there can be vectorial transmission, direct transmission (Rabies) and indirect transmission.

 In lines 444 - 449 the authors briefly explain the dilution effect, where they state that in conserved sites, the diversity of communities reduces the spread of zoonoses through the regulation of susceptible populations. However the degree to which biodiversity will regulate infection by a specific parasite is dependent on how deterministic host assembly is, whether the pathogen is niche- or dispersal-limited, and how increases in richness affect host and vector abundance. It is suggested that the concept be revised to provide a clearer explanation.

Reviewer #3: (No Response)

**Summary and General Comments**

Reviewer #1: After reading the manuscript I continue to go back to how these studies were selected and it was not hard to find a study through my own search which should have been included in the analysis. Only having 74 studies seemed low as soon as I read it and confirmed once I started looking for studies that should have been included.

Here are some examples: 

PMID: 37652628, PMID: 38060864, PMID: 36184108, PMID: 31593524...

Reviewer #2: It's an interesting study that offers a spotlight on the occurrence of pathogens and their relationship to land use change on the Yucatan Peninsula. It is obvious that one of the most common issues in this type of review is publication bias, but it is something that can be explained and justified. To have a better understanding of the results, a more detailed description of the differences between the region's states is recommended. It would have been interesting to debate or report on articles in which they searched for a pathogen other than the reported but found negative results.

Reviewer #3: L30-33. Please revise the sentence. It looks like a conclusion.

L40. Revise the use of the word competent. If the authors consider that a competent zoonotic agent is correct, they should include the data and the analysis to prove parasite competence.

L42-45. Please revise the sentence. The tone of the sentence is inappropriate for a revisión manuscript. Someone would use that sentence in an experimental or field study conclusion.

L45-46. Again, a review manuscript should not conclude on the observations of the studies. Should conclude about tendencies or common results. The paragraph states that landscape structure and conditions favor the prevalence of different zoonotic diseases; however, it contradicts the data shown in the manuscript. The evidence showed that landscape structures display different prevalences between the diseases but are not meant to favor the prevalence. 

L48-50. Please revise the sentence. The authors should estimate zoonotic disease incidence to support the argument.

L53-54. The socio-ecosystem perspective is interesting; however, the manuscript should use this approach to conclude it because the manuscript is "one health" oriented. Please provide the socio-ecosystemic framework in

---

## [Editor Report · Decision Letter 1]

11 Jun 2024

Dear DR VAZQUEZ-DOMINGUEZ,

We are pleased to inform you that your manuscript 'Prevalence and transmission of the most relevant zoonotic and vector-borne pathogens in the Yucatan peninsula: a review' has been provisionally accepted for publication in PLOS Neglected Tropical Diseases.

Best regards,

Angela Monica Ionica, Ph.D.

Academic Editor

Justin Remais

Section Editor

---

## [Editor Report · Acceptance letter]

26 Jun 2024

Dear Dr. Vázquez-Domínguez,

We are delighted to inform you that your manuscript, "Prevalence and transmission of the most relevant zoonotic and vector-borne pathogens in the Yucatan peninsula: a review," has been formally accepted for publication in PLOS Neglected Tropical Diseases.

Best regards,

Shaden Kamhawi

co-Editor-in-Chief

Paul Brindley

co-Editor-in-Chief
